# The Microbiota and Equine Asthma: An Integrative View of the Gut–Lung Axis

**DOI:** 10.3390/ani14020253

**Published:** 2024-01-13

**Authors:** Laurence Leduc, Marcio Costa, Mathilde Leclère

**Affiliations:** 1Clinical Sciences Department, Université de Montréal, Saint-Hyacinthe, QC J2S 2M2, Canada; laurence.leduc.1@umontreal.ca; 2Veterinary Department of Biomedical Sciences, Université de Montréal, Saint-Hyacinthe, QC J2S 2M2, Canada; marcio.costa@umontreal.ca

**Keywords:** heaves, horse, inflammatory airway disease, microbiome, recurrent airway obstruction

## Abstract

**Simple Summary:**

Ex vivo, mouse, and human studies have pointed to the gut microbiota as playing roles in many diseases, including asthma susceptibility and severity. Equine asthma shares many similarities with human asthma, and gut microbiota could also be critical components in the pathophysiology of the disease. The purpose of this review was to describe the current knowledge on the potential role of the understudied gut–lung axis in the pathophysiology of equine asthma.

**Abstract:**

Both microbe–microbe and host–microbe interactions can have effects beyond the local environment and influence immunological responses in remote organs such as the lungs. The crosstalk between the gut and the lungs, which is supported by complex connections and intricate pathways, is defined as the gut–lung axis. This review aimed to report on the potential role of the gut–lung gut–lung axis in the development and persistence of equine asthma. We summarized significant determinants in the development of asthma in horses and humans. The article discusses the gut–lung axis and proposes an integrative view of the relationship between gut microbiota and asthma. It also explores therapies for modulating the gut microbiota in horses with asthma. Improving our understanding of the horse gut–lung axis could lead to the development of techniques such as fecal microbiota transplants, probiotics, or prebiotics to manipulate the gut microbiota specifically for improving the management of asthma in horses.

## 1. Introduction

Equine asthma is a chronic and complex disease characterized by airway inflammation, mucus hypersecretion, and bronchoconstriction in response to inhaled antigens. It affects up to 15% of adult horses in its severe form and can greatly impact a horse’s athletic performance and quality of life [1,2]. Inhalation of environmental allergens such as antigens from mites and fungi, pollens, and endotoxins can trigger clinical exacerbations. During these exacerbations, airway hyperresponsiveness (AHR) leads to respiratory obstruction and clinical signs such as increased respiratory rate and effort, cough, and wheezing [3,4]. Although the role of aerosolized antigens in the pathophysiology of equine asthma exacerbations is well established, the factors contributing to the development and persistence of the disease remain largely uncertain. Recent research in rodents and humans has highlighted the significance of the gut–lung axis in the development of asthma [5,6,7]. 

In recent years, advancements in next-generation sequencing and bioinformatics have greatly enhanced our knowledge of equine gastrointestinal microbiota [8,9,10]. However, due to significant intra- and inter-host variability, it remains challenging to establish clear criteria for defining healthy microbiota. However, it is now understood that functional gut microbiota play a variety of physiological roles, such as producing short-chain fatty acids (SCFAs), enhancing local mucosal immunity, and promoting immunotolerance [6]. Fecal microbiota are greatly influenced by an individual’s diet and environment, and the intestinal microbiota of horses with and without asthma adapt differently to environmental changes, highlighting the possible role of altered gut microbiota in asthma [11,12]. In this review, the terms microbiota, referring to the microbes colonizing body surfaces, and microbiome, defined as the genome of these microbes, were employed according to their uses in the original references cited. Although microbiota include fungi, parasites, and viruses, this review mainly focused on bacterial microbiota.

The literature on the gut–lung axis and host–microbial community interactions in horses is sparse as current research on microbiota in horses is mostly descriptive and focuses on either the gut or the respiratory microbes separately [13]. Addressing the causality conundrum of the chicken or the egg concerning the role of the microbiota in equine asthma was a challenge. This review aimed to evaluate the potential impact of the gut–lung axis on the development and persistence of equine asthma. In this article, we first summarize the major determinants in the pathogenesis of horses and humans with asthma. We then discuss the gut–lung axis and propose an integrative view of the interactions between gut microbiota and asthma. Finally, we review therapies targeting the modulation of gut microbiota in horses. 

## 2. Methodology

Publications were selected if they were in English and reported original research or reviewed literature on asthma or gut microbiota in horses. Studies on humans and laboratory animals and ex vivo studies were included if the results were relevant. The following keywords were used to conduct most of the literature searches using different databases (e.g., Pubmed, ScienceDirect, and Scopus): mild to moderate and severe equine asthma, inflammatory airway disease, heaves, recurrent airway obstruction, equine gut microbiota, equine gut microbiome, equine gut dysbiosis, equine probiotics, gut–lung axis, asthma, and dysbiosis. The most recent literature search was carried out in December 2023.

## 3. Determinants in the Pathogenesis of Asthma: A Focus on the Impact of Gut Microbiota

### 3.1. Role of the Environmental Microbiome

#### 3.1.1. Inhaled Antigens as Causative Agents

The environment is a key factor in the development of equine asthma exacerbations, as demonstrated by the favorable response of affected horses to hay avoidance [2,14]. Horses that are kept in stables are cumulatively exposed to high levels of aerosolized particles such as fungi, endotoxins, mites, inorganic dust, and beta-D-glucans [15,16,17]. In young Thoroughbred and Standardbred racehorses, both tracheal mucus and bronchoalveolar lavage fluid (BALF) eosinophilia and neutrophilia are associated with respirable particle exposure [18,19,20]. Furthermore, older horses with asthma develop BALF neutrophilia when exposed to high levels of dust extracts and endotoxins [21]. These findings support the evidence that inhaled antigens from the environment are key factors in the pathophysiology of asthma. The impact of other environmental factors on airway inflammation such as exposure to pollutants, variations in environmental temperature, and pollen are also likely contributors to the disease severity but to a lesser extent [19,22,23].

#### 3.1.2. Antigens as Protective Agents

Inhaled antigens play a role in the initiation of asthma exacerbations, but exposure to antigens at an early age may also be protective against the disease. According to Strachan’s hygiene hypothesis, as proposed in 1989, the escalating incidence of allergic diseases such as asthma could be, in part, caused by reduced exposure to environmental antigens that have resulted from higher standards of cleanliness and limited contact with animals [24]. In a mouse model of allergic asthma, the intranasal administration of grass arabinogalactan, an extract from cowshed dust, which is a source of immuno-modulating substances, prevented mice from developing allergic airway inflammation and AHR [25]. Studies evaluating the incidence of asthma in children raised on farms or in urban settings have also highlighted the protective effect of the environment on the development of allergic diseases [26,27,28]. Supporting Strachan’s hygiene hypothesis, hay fever and atopic asthma in children are inversely correlated with endotoxin (a bacterial wall component) concentrations in their mattresses [29]. The environmental microbiome could even have an impact on asthma susceptibility before birth as offspring from pregnant mice exposed to a cowshed-derived bacterium were protected from asthma [30]. Similar findings were observed in humans as maternal exposure to an agricultural environment was associated with the increased protection of the children against allergic illnesses such as asthma [31,32]. These studies and others support the following two main hypotheses that are not mutually exclusive: early antigen exposure could induce immune tolerance to those antigens later in life, and exposure to a microbe-rich environment could make the gastro-intestinal, skin, and respiratory microbiota richer and more diverse, which has indirect and, in general, beneficial effect on a host’s immune system, which is discussed later in this article. Whether exposure to stables with higher levels of aerosolized particles, microbe-rich environments, or use/overuse of antimicrobials during pregnancy or early in life have protective or detrimental effects on asthma susceptibility in horses has not been investigated. Nevertheless, one abstract reported that there is a significant association between the diversity of gut bacteria at one month of age and the risk of several adverse health outcomes later in life [33].

### 3.2. Role of Airway Microbiota and Pathogens

The lungs have traditionally been thought of as a sterile environment. The presence of a respiratory tract microbiome is now well-recognized, in part due to the development of next-generation sequencing [34]. In horses, tracheal and pulmonary microbiota have been studied with culture-free methods, such as qPCR and next-generation sequencing [35,36,37]. Lower respiratory tract microbiota are thought to originate from the oropharynx following mucosal dispersion and micro-aspirations, which result in lower airway diversity and richness compared to the upper airways in humans and horses [32,36,38,39]. These findings have contributed to the hypothesis that, rather than microbial persistence and growth within the lower respiratory tract, the bacterial microbiota in the lungs of healthy individuals are determined by the state of balance between the migration of bacteria from the upper airway and their removal [40]. The role of respiratory microbes in preserving a healthy lung milieu, notably by regulating the immune system and preventing the spread of respiratory pathogens, is now acknowledged. As in other systems, they can produce metabolites, compete with potential pathogens, and contribute to maintaining homeostasis (Figure 1) [41]. To date, some mechanistic studies have demonstrated the relative importance of the airway microbiota in maintaining a healthy lung environment. One of them showed that human-derived oral commensal bacteria administered in the tracheas of mice induced T-helper cell type 17 responses and increased resistance to a common bacteria pathogen (*Streptococcus pneumoniae*) [42]. Furthermore, altered respiratory microbiota have been proposed as the cause of the persistence and perpetuation of airway inflammation [43]. 

In horses, most data have been observational. The lower airway microbiota of healthy horses and horses with asthma markedly differ from one another according to recent studies [35,37]. Manguin et al. found that sport horses with asthma had decreased abundances of the commensal bacteria *Corynebacterium* spp. as well as lower overall bacterial loads based on 16S rRNA gene qPCR [35]. This suggested that bacterial overgrowth is not a prominent feature of asthma in middle-aged horses, which contradicted past findings in racehorses, which are typically younger. The association between bacterial and viral pathogens, tracheal mucus, tracheal inflammation, and respiratory disease in racehorses has also been observational [44,45,46]. Environmental contamination and increased micro aspirations during strenuous exercise could play roles in mild-to-moderate asthma, as could common viruses such as equine rhinitis and herpes, but it is difficult to conclude on their contributions since they are ubiquitous [47,48]. Furthermore, the findings of past studies investigating lower airway microbiota and asthma need to be interpreted with caution as the environment and corticosteroid therapy both affect lower airway microbiota compositions in horses. Indeed, the tracheal microbiota in healthy and asthmatic horses are modified following systemic and nebulized dexamethasone administration [36]. 

In humans, asthma severity, prevalence, phenotype, AHR, and response to treatment are associated with airway dysbiosis. For example, certain bacteria (i.e., *Moraxella)* have been associated with increased abundances of neutrophils in the sputum samples of subjects with asthma, and a reduced response to corticosteroid therapy is linked to the presence of *Haemophilus* [49,50,51]. The influence of airway microbiota on asthma susceptibility may start very early in life. Upper airway microbes are acquired at birth, and the mode of delivery influences the composition and richness of the respiratory microbial populations. This could explain why offspring born via Caesarian section are at higher risk of developing asthma [52,53]. The influence of the mode of delivery on microbiota is further discussed in Section 4.1. The respiratory microbiota diversity and composition during the first months of life may have a significant impact on an infant’s susceptibility to asthma. Bisgaard et al. showed that the presence of certain bacteria such as *Streptococcus pneumoniae* in the hypopharynx of one-month old babies could predict diagnoses of asthma 5 years later [54]. Similarly, the nasopharyngeal colonization of babies with *Streptococcus pneumoniae* along with *Haemophilus influenzae* and *Moraxella catarrhalis* has been associated with asthma at 7 years of age [55]. Therefore, the importance of the respiratory microbiome at a young age has been highlighted in human medicine, but long-term longitudinal studies on horses are rare [33]. Since Caesarean sections are uncommon in mares, it is unlikely that a potential effect of the mode of delivery on the prevalence of equine asthma could be studied in a timely manner. However, other factors that could influence respiratory microbiota diversity at an early age could play roles, such as being born in a crowded barn versus in a field, being exposed to other foals, and being exposed to antimicrobials during pregnancy and in the first few months of life. Furthermore, the extent to which the fecal microbiota can impact the vaginal microbiota and, therefore, a foal’s colonization deserves better investigation [56]. Wild horses and horses receiving forage-only diets have higher diversity and different compositions [57,58]. 

### 3.3. Impact of Lifestyle on Microbiota and Asthma Susceptibility

#### 3.3.1. Diet and Obesity

One of the ways digestive health could influence respiratory health and, specifically, asthma prevalence and severity, could be via poor diet, altered microbiota, and obesity. Studies in the human literature suggest that poor diet quality, obesity, and sedentary lifestyles increase asthma susceptibility, worsen prognosis, and influence asthma phenotypes [59,60]. In addition to the mechanical interference of adipose tissue with the movements of the rib cage and diaphragm, the link between obesity and asthma could be, in part, explained by leptin, an inflammatory mediator secreted by adipose tissue that promotes T cell proliferation and activation, as well as macrophages recruitment [61]. In lean mice sensitized with ovalbumin (OVA), AHR was enhanced following leptin infusion, and microbiota-depleted mice had enhanced leptin sensitivity [62,63]. Dysbiosis could, therefore, theoretically increase leptin sensitivity, and its inflammatory effects could promote AHR. Leptin levels are increased in overweight horses, but leptin’s association with asthma has not yet been studied [64]. Not only does obesity increase the likelihood of developing asthma, but it also makes the control of the disease more challenging. When evaluating the relationship between body mass index and response to fluticasone (an inhaled corticosteroid) with or without salmeterol (a long-acting β agonist), Boulet and Franssen found decreased likelihoods of achieving asthma control in class 3 obese patients (6% of those receiving fluticasone or a combination of fluticasone and salmeterol) when compared to lean patients (78% of those receiving the same treatments) [65]. The link between the gut microbiome and obesity is described in the next paragraphs. 

The literature evaluating the association between asthma and obesity in horses is sparse. Limited evidence has suggested that obesity is a risk factor for equine asthma [66,67]. This is particularly interesting in the face of the proposed mechanism described above because obesity in horses is often a consequence of inadequate exercise and “poor diet”, such as high-energy diets rich in non-structural carbohydrates and low in soluble fibers. However, mild and moderate asthma are frequent in performance horses and racehorses who are typically not obese. The link between obesity, microbiota, and equine asthma remains unexplored, for now, but the association between obesity, endocrine diseases, and gut microbiota compositions in horses has been investigated. Biddle et al. found that both richness and diversity were increased in obese horses whereas Elzinga et al. identified decreased diversity in horses affected by equine metabolic syndrome (i.e., a predisposing factor to obesity) [68,69]. A possible explanation for the conflicting results may have stemmed from the fact that diet was not controlled in either study. In a diet-controlled study with a more homogenous population, fecal microbiome diversity and Bacteroidetes abundance were increased in obese horses [70]. In addition, murine macrophages exposed to fecal extracts from obese horses exhibited increased expressions of inflammatory markers such as IL-1β, TNF-α, and IL-6 when compared to those exposed to fecal extracts from non-obese horses [71]. 

Diet and the gut microbiome could also contribute to the connection between obesity and asthma. As they are usually low in soluble fibers, obesogenic diets are associated with altered microbiomes, decreased metabolites production by gut bacteria, such as SCFA which have anti-inflammatory and immunomodulatory properties [72]. Major SCFA producers such as Bacteroidetes are reduced in obese patients [73]. The impact of low- and high-fiber diets on SCFA concentrations and asthma was elegantly shown in mice, where low concentrations of propionate (SCFA) were associated with increased allergic airway inflammation levels and AHR [7]. Short-chain fatty acid administration (i.e., propionate and acetate) also attenuated or inhibited the development of allergic airway inflammation in this study. Microbiota metabolites and diet could, therefore, both contribute to the dysregulation of inflammatory homeostasis occurring in asthma and in gut–lung axis crosstalk (detailed in Section 3). Notably, the relative abundance of *Fibrobacter*, an SCFA-producer, was increased in healthy horses eating hay but not in horses with asthma on the same diet, which suggested that this crosstalk could also apply to horses [12]. However, SCFAs were not measured in that study, and so this remains speculative.

#### 3.3.2. Exercise

Although it is clear that a lack of physical activity can contribute to human obesity and, consequently, to asthma, increased asthma prevalence and worsened asthma control have been associated with physical inactivity even in non-obese patients, suggesting that other factors are involved [74]. Improvements in AHR and cellular airway composition have been shown with regular physical activity, though the exact mechanisms by which asthma outcomes are improved are not fully understood [75]. Interestingly, decreases in eosinophils and total cells in the sputum and BALF of humans and ovalbumin-sensitized mice, respectively, have been observed following exercise [76,77,78]. Metabolomic pathways can also be affected by physical activity, with high concentrations of butyrate being associated with good cardiorespiratory fitness and mitigation of the negative impacts of a high-fat diet on the gastrointestinal microbiome [79,80].

The association between asthma and physical inactivity has not been investigated in horses, although there is limited evidence suggesting that intense training can transiently modify the gut microbiota composition [81]. In one study, blood metabolomics (including alanine and valine) before an endurance race were associated with gut microbiota but not with performance [82]. However, the 1H nuclear magnetic resonance approach used in this study only detected metabolites with high concentrations, and the interpretation of metabolite peaks can be ambiguous with this technique. The stress associated with intense exercise could also induce lower-airway inflammation as stress-related behaviors (touching a rubber tie-cord) are correlated with tracheal inflammation [83]. Yet, such behaviors are also correlated with a decreased frequency of head lowering, which may also affect tracheal inflammation. Because moderate exercise following transport increases intestinal permeability and systemic inflammation biomarkers (i.e., serum amyloid A and lipopolysaccharide) in horses, it can be hypothesized that bacterial translocation from the gastrointestinal tract to the respiratory tract could result in lower airway inflammation and/or be involved in asthma pathophysiology [84].

#### 3.3.3. Antibiotic Exposure 

Antibiotic exposure is yet another variable affecting the microbiome, and consequently, it has the potential to increase asthma susceptibility. Microbiome composition modifications following antimicrobial administration in horses are now being recognized. For example, fecal microbiota were modified shortly after the initiation of antibiotic treatment in healthy horses, and the alterations in the bacterial communities took at least 25 days to recover from [8]. Bacterial species diversity and richness were significantly decreased following trimethoprim sulfadiazine administration [8,85], and the relative abundance of Bacteroidetes decreased after ceftiofur administration [86]. A decrease in SCFA-producing bacteria such as Bacteroidetes could increase asthma susceptibility, but to date, convincing evidence that the fecal microbiota modifications observed following antibiotic administration in horses contributes to airway inflammation is lacking. 

In humans, a systematic review by Baron et al. concluded that there is a moderate amount of evidence for an association between early life exposure to antibiotics and childhood asthma [87]. Furthermore, in a cohort of 143,000 children, asthma was associated with antibiotic administration in the first year of life [88]. In agreement with those findings, early alterations in the microbiome following antibiotic administration can also affect immune function and IgA responses, which are associated with an increased susceptibility to human allergic diseases [89]. Dysbiosis resulting from antibiotic administration therefore appears to increase susceptibility to asthma, but mostly later in life. This could explain why exacerbations are not typically observed clinically immediately following antibiotics administration, in horses with asthma. 

#### 3.3.4. Sex

Before puberty, asthma is more prevalent in boys. However, after puberty, asthma is more common in females, and women experience more asthma-related morbidity and mortality. The reasons for this are likely multifactorial, but sex hormones appear to play a role, as testosterone appears to have a protective effect against asthma [90]. In horses with asthma, there have been reports of a predisposition for mares [91], but this is controversial [1]. Applying what we know from human medicine, we can hypothesize that the protective effect of having more testosterone is lost in male horses because most males are geldings. This could explain why differences are not consistently observed between males and females. According to Mshelia et al., there were significant differences in the fecal microbiome of mares and stallions [92]. However, the study did not include any geldings. In mice, the microbiome may contribute to sex differences observed in airway hyperresponsiveness, as these sex differences disappear when the mice are treated with antibiotics to ablate the gut microbiome [93]. In humans, airway microbiome differ between males and females, and there is an association between airway microbial markers, asthma, and sex [94]. However, to the authors’ knowledge, there are no known sex-specific patterns in the gut microbiota of humans associated with asthma.

## 4. Gut–Lung Axis

Host–microbe interactions can exert impacts beyond their local environments and influence immunological responses in remote organs. Both microbe–microbe and host–microbe interactions can have long-reaching effects, and the crosstalk between the gut and the lungs is defined as the gut–lung axis. The gut–lung axis concept is supported by complex connections and intricate pathways involving both the gut and lung microbiota [95]. As of now, most recognized pathways are in the gut-to-lung direction.

### 4.1. Short-Chain Fatty Acids: Chemical Messengers

Short-chain fatty acids such as acetate, butyrate, and propionate are produced through the fermentation of fibers by fibrolytic bacteria such as members of the phyla Bacteroidetes and Fibrobacter. They regulate the barrier function of the gut by stimulating intestinal epithelial cells to secrete mucus and antimicrobial peptides, and they also upregulate tight junction proteins [96]. SCFAs can also attenuate inflammatory and allergic responses in the lungs by communicating with pulmonary antigen-presenting cells [96]. For example, diets with high-fiber contents increase circulating SCFAs, which modulate dendritic cell function in the lungs, as was demonstrated in a mice model of asthma [7]. In that study, propionate administration enhanced dendritic cells’ phagocytic activities and decreased their capacity to induce T2 inflammation responses. SCFAs can also alter cytokine and chemokine production and inflammatory cell proliferation and affect local and systemic immunity by promoting regulatory T cells (Tregs) [96]. The role of SCFAs in local and systemic immunity is illustrated in Figure 2. 

### 4.2. Regulatory T Cells and T2 Responses

Tregs play central roles in antigen tolerance and immune homeostasis, especially in allergic diseases. The production of local and systemic mediators by gut anaerobic bacterial fermentation, primarily SCFAs, can regulate the generation of Tregs, which modulate T2 responses. Germ-free mice colonized with microbiota of lower diversity were more susceptible to developing T2 responses, atopy, and asthma [6,97]. Such increased T2 activity can lead to a hyperreactive response against commensal bacteria, which is normally considered inoffensive. The increased mRNA expression of both IL-4 and IL-5 in the pulmonary lymphocytes of asthmatic horses suggested that the dysregulation of T2 responses contributed to the pathophysiology of the disease [98].

The role of Tregs is not limited to T2 response modulation, and it includes the regulation of mucosal antibody production (i.e., IgA). Tregs reduce systemic inflammation and CD4+ T cell activation by modulating the secretion of IgA to eliminate microbial ligands [99]. The proportion of Tregs depends on the gut microbiota composition, as members of the genera *Clostridium*, *Lactobacillus*, and *Bifidobacterium* enhance their proliferation [100]. Theoretically, gut dysbiosis could alter the Tregs’ regulation of IgA secretion and contribute to asthma susceptibility. For example, lower levels of IgA-bound bacteria in children increased the risk of developing asthma [89].

### 4.3. Innate Lymphoid Cells and Mucosal Immunity 

Innate lymphoid cells (ILCs) lack typical lymphocyte surface markers but can release cytokines and express genes such as T helper cells. They are particularly abundant in the gut and lung mucosa and have receptors for cytokines released from damaged tissues [101]. ILCs play a crucial role in mucosal immunity, notably, by inhibiting viral and bacterial infections through the secretion of interferon gamma (IFN-y) and IL-22 [102]. Through MHC (major histocompatibility complex)-II mediated antigen presentation, ILCs also contribute to antigen tolerance. The gut microbiome can promote IL-22 secretion by ILC3s, and this induces the production of antimicrobial peptides by the epithelium and enhances epithelial barrier integrity. However, when inappropriately activated, certain lineages such as ILC2s can have detrimental effects and induce allergic diseases [102]. ILC2s can initiate and promote allergic airway inflammation by stimulating the secretion of IL-13 and the T2 differentiation of dendritic cells [103]. The proportion of ILC2s was increased in the peripheral blood samples of asthmatic adults when compared to controls, and it could be used as a biomarker to predict eosinophilic airway inflammation [104]. Not only are there increased numbers of ILC2s in patients with asthma, their functions and reactivity were also altered. For example, the ILC2s from patients with asthma secreted more IL-5 and IL-13 compared to the ILC2s derived from healthy controls [105]. Interestingly, ILC2s can be recruited from the gut and migrate to the lungs in response to inflammation and gut dysbiosis [106].

## 5. Integrative View of Gut Dysbiosis and Asthma

### 5.1. Gut Dysbiosis and Asthma in Humans

The hygiene hypothesis is supported by an association between early exposure to environmental antigens and reduced asthma susceptibility [107,108]. A theory called the ‘microbiota hypothesis’ (originally, the microflora hypothesis) has recently emerged and suggests that alterations in gut microbiota occurring early in life can promote allergic diseases and asthma by depleting the microbial communities responsible for immunological tolerance [100,109]. The period of life in which alterations in microbial communities can promote the later development of diseases is called the ‘window of opportunity’. 

Evidence of gut dysbiosis, illustrated by an increase in *Clostridium* spp. and a decrease in *Lachnospira* spp., was observed in a population of asthmatic children [110]. Similarly, colonization by *Clostridium* (now *Clostridioides*) *difficile* at one month of age was associated with asthma in childhood [111]. Another study showed that decreases in the relative abundances of *Lachnospira* and fecal SCFAs (acetate) in 3-month-old infants were associated with increased risks of developing childhood asthma [112]. The same group found no significant differences in the gut microbiota compositions between older children with asthma and atopy and healthy children [113]. These results highlighted the importance of the ‘window of opportunity’ in young infants for modulating susceptibility to allergic diseases. As discussed in Section 2, the mode of delivery can impact respiratory microbiota in newborns. There is also evidence that it can affect gut microbiota and thereby influence asthma susceptibility [53]. Stokholm et al. found that babies born via Caesarian section were at increased risk of developing asthma if their gut microbiota compositions remained similar to their microbiota profiles at birth [114]. While Caesarean sections are associated with decreased abundances of and diversity in Bacteroidetes and increased abundances of Firmicutes during the first three months of life, the impact of mode of delivery on gut microbiota colonization and diversity appears to lessen after 6 months [53]. While these studies illustrated the effects of early gut microbiota disturbances on asthma susceptibility, the minimal proportion of birth via Caesarean section in horses and the high prevalence of severe equine asthma (15%) suggests that other factors predominate. 

The fecal microbiota of asthmatic patients differ from those of healthy subjects. For example, a relationship between allergen sensitization and fecal microbiota structure with decreased Bacteroidetes: Firmicutes ratios was observed in patients with asthma [115]. Beyond alterations in gut microbiota composition, its metabolites are also modified in asthma. Significant decreases in fecal SCFAs, including acetate, propionate, and butyrate, were detected in patients with asthma in two studies [116,117]. Gut microbiota can also have an impact on lung function in other diseases. For instance, in patients with chronic obstructive pulmonary disease (COPD), worsening lung function was also associated with lower fecal Firmicutes while stable function was associated with higher fecal Bacteroidetes [118]. In another study, *Prevotella* was overrepresented in a cluster of patients with reduced lung function [115]. However *Prevotella* relative abundances were reduced in asthmatic patients in another study, [119]. The exact pathways by which gut microbiota influence asthma susceptibility and persistence remain uncertain (Figure 3). 

### 5.2. Gut Dysbiosis and Asthma in Horses

The current research on microbiota in horses is mostly descriptive and has focused on either gut or respiratory microbes separately. To the authors’ knowledge, there is only one study investigating both the gut microbiota and asthma in horses. Leclere and Costa found that the intestinal microbiota differed between healthy and asthmatic horses [12]. Healthy horses transitioning from pasture to hay diets had increases in fecal Fibrobacter, and this was not observed in the horses with asthma. While differences were observed between the asthmatic horses in remission and the controls, they were less marked, which may suggest that gut microbiota alterations mostly occur in exacerbation. Some of the differences observed between the controls and the asthmatic horses in that study were similar to those seen in adult humans with asthma. For example, 8 of the 15 overrepresented genera in the horses with asthma belonged to the phylum of Firmicutes. *Prevotella* was also increased in horses in exacerbation compared to the controls. Because the sample size of this study was small and asthma exacerbation is inherently associated with diet change, it is difficult to conclude if the gut microbiota changed due to the disease or to the diet modifications. Evaluating the fecal microbiota in horses in which exacerbation was provoked without modifying the diet would greatly improve our understanding of the causality between the gut microbiome and equine asthma.

## 6. Microbiota-Directed Therapies and Modulation of the Gut–Lung Axis

With the increasing knowledge on how microbiota can contribute to asthma, strategies to restore microbial homeostasis have gained growing interest. However, the literature evaluating the efficacy of microbiota manipulation techniques in horses is sparse, and it outlines inconsistent results. In this section, we summarize the different techniques for gut microbiota manipulation including prebiotics, probiotics, and postbiotics. The efficacy of those techniques for the prevention or treatment of equine asthma has not been evaluated, and therefore, a few examples from human medicine are listed. 

Prebiotics are substrates utilized by host microorganisms that exert benefits for the host. The most commonly used prebiotics are oligosaccharides, such as fructo-oligosaccharides (FOS) or mannan-oligosaccharides (MOS) [120]. They can prevent colonization by pathogens, stimulate the growth of probiotics, and undergo fermentation, resulting in the increased production of SCFAs [121,122,123]. Studies evaluating the effects of prebiotics on equine gut microbiota are limited, and the ones evaluating their impact on asthma are simply lacking. The effects of an oligosaccharide-rich diet on pregnant mares and their foals were investigated by Lindenberg et al., and they found that the supplemented foals had significantly higher relative abundances of *Akkermansia* spp. [124]. Interestingly, *Akkermansia mucinophila* abundance was decreased in the guts of children with allergic asthma [125]. While a four-week treatment with symbiotics (prebiotics and probiotics) did not alter bronchial inflammation in human patients with asthma, significant decreases in the systemic production of T2-cytokines such as IL-5 were observed in one study [126]. A prebiotic containing galactooligosaccharides was administered for 3 weeks in adults with asthma in another study, and decreased AHR associated with hyperpnea-induced bronchoconstriction was recorded [127]. These results hint at a potential for prebiotics in asthma management.

Probiotics are referred to living microorganisms with the capacity to restore microbial imbalances by preventing colonization by pathogens [120]. By upregulating tight junction proteins, they can enhance gut barrier integrity. They also have immunomodulatory properties by regulating the expression of Treg cells and decreasing T17 responses [121,128]. Developing effective probiotics for horses is challenging because the ideal healthy microbiota have not been determined. Currently published studies do not provide conclusive evidence of their benefits and yield contradictory results. Yeast probiotics have various advantages over probiotics containing mainly bacteria, such as their resistance to acidic environments and antimicrobials, which are often used in ill patients with concomitant dysbiosis [129]. *Saccharomyces cerevisiae* or *boulardii* are two nearly identical strains of non-pathogenic yeasts that can release proteases to degrade *C. difficile* toxins A and B [129]. The effects of Saccharomyces in horses with asthma have yet to be examined, but in mice models of asthma sensitized to OVA, significant reductions in AHR and airway inflammation were observed in the mice treated with *S. cerevisiae* [130,131]. 

In foals, probiotics may be useful in modifying their gut microbiota compositions because the ‘window of opportunity’ to permanently alter the gut microbiota is thought to be between birth and 50 days of age [132]. Alas, studies investigating the administration of probiotics in foals have not assessed the effects on microbiota compositions or asthma susceptibility, but rather, they primarily outline gastrointestinal clinicopathologic findings, which are beyond the scope of this review. In newborn mice, the administration of *Lactobacillus rhamnosus* and *Bifidobacterium lactis* during OVA sensitization and challenge suppressed airway reactivity and pulmonary eosinophilia [133]. In contrast, a meta-analysis of clinical trials assessing the effects of probiotic supplementation on atopy and asthma concluded that the evidence supporting their use in children to prevent asthma is currently insufficient [134]. It is important to note that probiotic administration in foals is considered generally safe, but it can lead to adverse effects, such as an increased incidence of diarrhea requiring veterinary intervention [135]. Therefore, the use of probiotics in foals to alter the gut microbiota requires further investigation.

Postbiotics are soluble products and metabolites secreted by gut microbial communities known for their protective effects on intestinal epithelium, immunomodulation functions, and selective cytotoxicity against tumors. The most well-known example of postbiotics are SCFAs. Propionate-supplemented water was given to mice sensitized to house-dust-mites in a model of allergic asthma [7]. Inflammatory cellular infiltration was reduced in the airways of the supplemented mice, and overall inflammatory responses were also decreased. Likewise, children were less likely to have asthma between 3 and 6 years of age if their fecal levels of butyrate and propionate were high [136]. These findings suggest that postbiotics such as SCFAs may decrease asthma susceptibility, but this awaits further clarification. 

To the authors’ knowledge, fecal microbiota transplants have not been investigated for the treatment or prevention of either equine or human asthma. 

## 7. Conclusions

The purpose of this review was to describe the role of the understudied gut–lung axis in the pathophysiology of equine asthma. Ex vivo, mouse, and human studies have pointed to gut microbiota as important components in asthma susceptibility and severity. While equine asthma shares many similarities with human asthma, further research is certainly needed to understand the implications of gut microbiota compositions and functions on equine asthma. The horse has the potential to serve as a model for human asthma across its lifespan. By focusing on the large-scale data integration of longitudinal equine health records from foals to adulthood, researchers could investigate key aspects of the effects of disease and antimicrobial administration during the “window of opportunity” on equine asthma susceptibility. If promising trends emerge, the horse could serve as a model for microbiota manipulation during the susceptible “window of opportunity” to reduce the risk of developing asthma in adulthood. Improving our understanding of the horse’s gut–lung axis could also lead to the development of techniques to manipulate the gut microbiome for the treatment of asthma in horses. 

## Figures and Tables

**Figure 1 animals-14-00253-f001:**
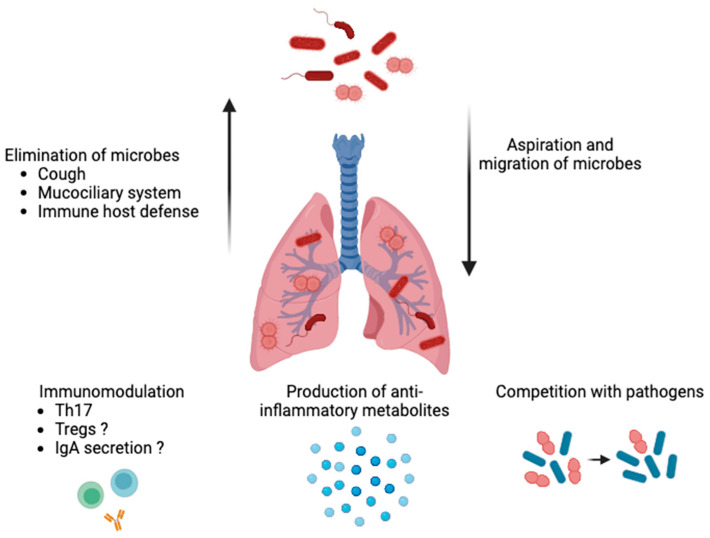
Role of airway microbiota in healthy lungs. The bacterial microbiota are determined by the state of balance between the micro-aspirations from the upper airways and migration of bacteria along the mucosal surfaces and their clearance (e.g., via coughing, the mucociliary system, and a host’s immune defenses). Respiratory microbiota can produce metabolites, compete with potential pathogens for space and nutrients, contribute to maintaining homeostasis (e.g., pH and oxygen tension levels), and promote immunomodulation (by inducing a T-helper cell-type 17 response and, possibly, regulatory T cells (Tregs) and immunoglobulins (Ig) A). The image was created using BioRender.com, accessed on 16 December 2023.

**Figure 2 animals-14-00253-f002:**
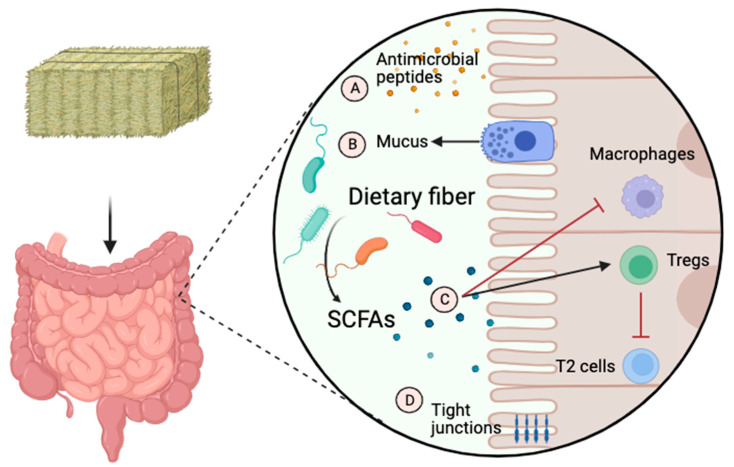
The role of short-chain fatty acids in local and systemic immunity. Short-chain fatty acids (SCFAs) such as acetate, butyrate, and propionate are produced through the fermentation of fibers by fibrolytic bacteria. They regulate the gut’s barrier function by stimulating intestinal epithelial cells to secrete antimicrobial peptides (A) and mucus (B). They attenuate inflammatory and allergic responses in the lungs (C) by communicating with regulatory T cells (Tregs) and antigen-presenting cells such as macrophages. They also upregulate tight junction proteins (D). T2, type 2 inflammation. The image was created using BioRender.com, accessed on 16 December 2023.

**Figure 3 animals-14-00253-f003:**
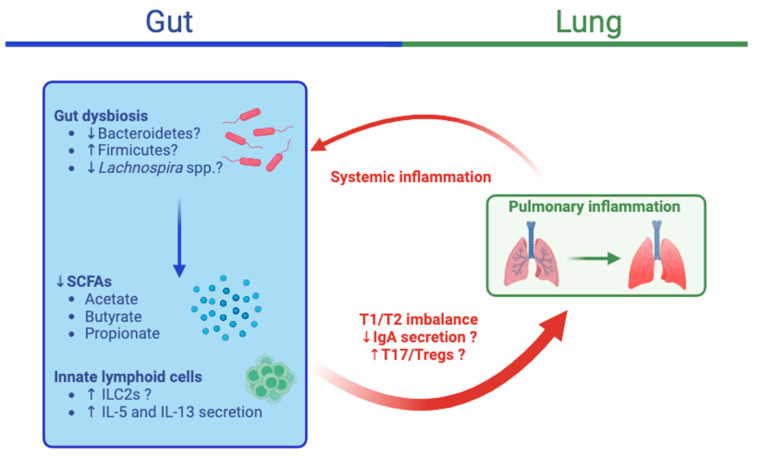
Integrative view of the gut–lung axis in asthma. Gut dysbiosis can lead to the decreased production of short-chain fatty acids (SCFAs) due to decreases in the relative abundances of Bacteroidetes and Lachnospira spp or an increase in Firmicutes. Furthermore, innate lymphoid cells 2 (ILC2s) can stimulate the secretion of interleukin (IL)-5 and IL-13 and the T2 differentiation of dendritic cells, which can initiate and promote allergic airway inflammation. The dysregulation of IgA secretion, T1/T2 imbalances, and increased T17/Treg responses resulting from these changes can lead to airway inflammation, as has been observed in asthma. The crosstalk from the lungs to the gut remains unclear, but resulting systemic inflammation could further contribute to gut dysbiosis and inflammation. IL, interleukin; T1, type 1 inflammation; T2, type 2 inflammation. The image was created using BioRender.com, accessed on 19 December 2023.

## Data Availability

No new data were created or analyzed in this study. Data-sharing is not applicable to this article.

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
