# Peer review of "The Microbiota and Equine Asthma: An Integrative View of the Gut–Lung Axis"

_animals, 2024, doi:10.3390/ani14020253_

Round 1
Reviewer 1 Report
Comments and Suggestions for Authors
Congratulations on an interesting well written review. The scope of review is well defined along with definitions of principal components. There are several issues to consider.
Ln 203 to 211. While this may be true of obese horses and I understand this section has been included for completeness, given that equine asthma is such a huge problem in racehorses and high performance horses who are not obese - this section appears out of place. Given that the chief issue of obese horses is the risk of laminitis rather than equine asthma, the emphasis of obesity on equine asthma is rather overstated.
I would disagree that the cause of equine obesity is rarely a consequence of a poor diet. It absolutely is. The authors should elaborate on what the causes of equine obesity are. Generally high energy diets rich in non-structural carbohydrates and inadequate exercise. Sounds very much like the cause of obesity in people (with a degree of genetic predisposition thrown in).
Ln 220. As [diets] are usually low in soluble fibers, obesogenic diets are associated with altered microbiome and decreased SCFA production... Reviewer's comment - Exactly - which for a horse is a 'poor diet' as indicated in my earlier comment.
Ln 246. The issue with asthma in horses is its effect on physically active horses and its impairment on performance. This association of exercise/training as a stressor on the gut microbiome needs to be explored in this section. Why would an association between asthma and physical inactivity in horses be investigated - it is a non-event.
There has been no consideration of the effect of stressors on the gut microbiome and/or lung microbiome. Please consider work undertaken by Padalino et al on stress response and lower airway contamination. High performance horses are subjected to a number of stressors. Consequently, these horses are also most affected by equine asthma.
Author Response
Congratulations on an interesting well-written review. The scope of review is well defined along with definitions of principal components. There are several issues to consider.
Answer: Thank you for taking the time to review this manuscript and for your constructive comments. Please find the detailed responses below and the corresponding corrections highlighted in the re-submitted files. Note that some sections highlighted in the revised manuscript are in response to the other reviewers.
Ln 203 to 211. While this may be true of obese horses and I understand this section has been included for completeness, given that equine asthma is such a huge problem in racehorses and high-performance horses who are not obese - this section appears out of place. Given that the chief issue of obese horses is the risk of laminitis rather than equine asthma, the emphasis of obesity on equine asthma is rather overstated. I would disagree that the cause of equine obesity is rarely a consequence of a poor diet. It absolutely is. The authors should elaborate on what the causes of equine obesity are. Generally high energy diets rich in non-structural carbohydrates and inadequate exercise. Sounds very much like the cause of obesity in people (with a degree of genetic predisposition thrown in).
Answer: Thank you for pointing this out. The paragraph was rephrased according to your comment (lines 231 to 232).
Ln 220. As [diets] are usually low in soluble fibers, obesogenic diets are associated with altered microbiome and decreased SCFA production... Reviewer's comment - Exactly - which for a horse is a 'poor diet' as indicated in my earlier comment.
Answer: You are right. Lines 231-232 now read: obesity in horses is often a consequence of inadequate exercise and “poor diet”, such as high-energy diets rich in non-structural carbohydrates and low in soluble fibers.
Ln 246. The issue with asthma in horses is its effect on physically active horses and its impairment on performance. This association of exercise/training as a stressor on the gut microbiome needs to be explored in this section. Why would an association between asthma and physical inactivity in horses be investigated - it is a non-event. There has been no consideration of the effect of stressors on the gut microbiome and/or lung microbiome. Please consider work undertaken by Padalino et al on stress response and lower airway contamination. High performance horses are subjected to a number of stressors. Consequently, these horses are also most affected by equine asthma.
Answer: The association between asthma and physical inactivity is evoked in this paragraph because the previous one discusses the link between asthma and physical inactivity in humans. We agree with the reviewer that mild and moderate asthma is mainly relevant as it impairs performance, but inactivity in horses with severe asthma is definitively a problem that warrants attention. Many horses with severe asthma become less active, and are eventually retired by their owners. The severity of exacerbations often worsens and it is most often attributed to the natural progression of the disease. However, the effect of a sedentary lifestyle warrants investigation, as maintaining some level of physical activity (between exacerbations) might increase the quality of life of horses with severe asthma. The work by Padalino et al. was added (lines 278-280) and the effect of exercise as a stressor is now discussed in lines 273 and 278 to 286.
Reviewer 2 Report
Comments and Suggestions for Authors
The authors provide an interesting review of the contemporary status of gut-lung axis and airway hypersensitivity-related disease pathogenesis and potential management. Reported studies are well described and represented. Application of equine AHS models are appropriately incorporated into the review. An area that is not included that might shed additional light is the investigation of equine dysbiosis following antimicrobial administration and development of disease exacerbation. It is well recognized that horses can be particularly susceptible to dysregulation of gastrointestinal flora following a variety of antimicrobial therapies. This is an area of historic and active investigation. Is there evidence from this work that among horses that develop antibiotic-associated colitis, those with a previous diagnosis of asthma have evidence of asthma-associated disease exacerbation (heaves affected horse going into crisis)? From a clinical specialist, this has not been observed and this seems like a starting point for the fundamental argument that gut-lung axis is the foundation for pulmonary dysbiosis. As stated by the authors, the overall level of current evidence regarding the equine gut-lung axis and equine asthma is currently low, this is an area where more data is needed to answer important questions. The report is well prepared, although data is limited, the authors have worked to report on available publications that relate to the topic of interest. Editing is needed throughout the document to further improve the clarity of writing, but overall a well prepared report.
Specific comments,
1. Title, The microbiota and equine asthma: an integrative view of the gut-lung axis
2. Abstract, line 18, This review aims to report the potential role
3. Line 50, the genome of these microbes,
4. Line 98, leads to a microbe rich environment
5. Line 113, results in lower airway diversity
6. Line 137, the listed references are [35,,37], either an adjustment on comma or addition of 36 is indicated
7. Line 267, associated with antibiotic administration
8. Line 323, there is inconsistency with how nomenclature and abbreviations are reported. Standard process is the name follow by parenthetical abbreviation, in this instance the abbreviation is written and followed by (interferon gamma). This format should be corrected and consistent throughout the report.
9. Line 368, a relationship between fecal microbiota or association, as written a relation is not correct.
10. Line 368 (end of line), allergen sensitization of lower airway Bacteroides
11. Line 394, as written focalized is not clear. Mostly descriptive and focused on either gut or lung would be more clear
Author Response
The authors provide an interesting review of the contemporary status of gut-lung axis and airway hypersensitivity-related disease pathogenesis and potential management. Reported studies are well described and represented. Application of equine AHS models are appropriately incorporated into the review. An area that is not included that might shed additional light is the investigation of equine dysbiosis following antimicrobial administration and development of disease exacerbation. It is well recognized that horses can be particularly susceptible to dysregulation of gastrointestinal flora following a variety of antimicrobial therapies. This is an area of historic and active investigation. Is there evidence from this work that among horses that develop antibiotic-associated colitis, those with a previous diagnosis of asthma have evidence of asthma-associated disease exacerbation (heaves affected horse going into crisis)? From a clinical specialist, this has not been observed and this seems like a starting point for the fundamental argument that gut-lung axis is the foundation for pulmonary dysbiosis. As stated by the authors, the overall level of current evidence regarding the equine gut-lung axis and equine asthma is currently low, this is an area where more data is needed to answer important questions. The report is well prepared, although data is limited, the authors have worked to report on available publications that relate to the topic of interest. Editing is needed throughout the document to further improve the clarity of writing, but overall a well prepared report.
Answer: Thank you for taking the time to review this manuscript and for your constructive comments. Please find the detailed responses below and the corresponding corrections highlighted in the re-submitted files. Note that some sections highlighted in the revised manuscript are in response to the other reviewers. As of now, there is no evidence that among asthmatic horses developing antibiotic-induced colitis, asthma exacerbation occurs. In humans, dysbiosis caused by antibiotic administration seems to increase asthma susceptibility, but only later in life, so this may be why clinically we do not observe exacerbations immediately following disease. A few sentences to that effect were added in lines 305 to 308.
Specific comments
Answer: Thank you for your comments. We have addressed all of them in the appropriate lines.
- Title, The microbiota and equine asthma: anintegrative view of the gut-lung axis
- Abstract, line 18, This review aims to reportthe potential role
- Line 50, the genomeof these microbes,
- Line 98, leads to a microberich environment
- Line 113, results in lower airwaydiversity
- Line 137, the listed references are [35,,37], either an adjustment on comma or addition of 36 is indicated
- Line 267, associated with antibioticadministration
- Line 323, there is inconsistency with how nomenclature and abbreviations are reported. Standard process is the name follow by parenthetical abbreviation, in this instance the abbreviation is written and followed by (interferon gamma). This format should be corrected and consistent throughout the report.
- Line 368, a relationship between fecal microbiota or association, as written a relation is not correct.
- Line 368 (end of line), allergen sensitization of lower airway Bacteroides
- Line 394, as written focalizedis not clear. Mostly descriptive and focused on either gut or lung would be more clear
Reviewer 3 Report
Comments and Suggestions for Authors
Thank you for this interesting and timely review. It is well-written and provides the reader with a good understanding of the current state of knowledge and potential developments to be further explored. Although this is a narrative review and this is not strictly required, it may be helpful to describe your method for literature search.
Given the United States National Institute of Health’s policy on considering sex as a biological variable, I am wondering if you can comment on the role of sex differences in the gut-lung axis and equine asthma. In humans, young boys are more likely than girls to develop asthma, but after puberty females are more likely than males to develop asthma. There is evidence of sex differences in the microbiome. Are the same patterns found in horses? Have sex-differences related to the gut-lung axis been studied in the horse or is this an area where more research is needed?
Line 143: Is there any evidence in racehorses that strenuous exercise alters the GI blood flow and tight junctions creating a leaky gut syndrome and that bacterial translocation from the GI to the respiratory tract is part of the asthma disease process in race horses?
Line 449: Given the findings of the study of Schoster et al (PMID:25903509) where probiotic administration to healthy foals resulted in a greater incidence of diarrhea requiring treatment than in control foals, perhaps a word of caution is necessary regarding probiotic administration to foals?
Line 471: I wonder if you could take the conclusions a bit further and say that the horse is a potential model for human asthma over the lifetime. With a focus on large scale data integration of longitudinal equine health records from foaling to adulthood, important questions related to the effects of illness and antimicrobial administration during the so-called “window of opportunity” on the later development of equine asthma could be investigated. And if interesting patterns emerge, maybe the horse could serve as a model for administration of pre-, pro-, and postbiotics during the vulnerable “window of opportunity” to mitigate the risk of developing asthma later in life.
Reference [33]: Please provide complete citation.
Author Response
Thank you for this interesting and timely review. It is well-written and provides the reader with a good understanding of the current state of knowledge and potential developments to be further explored. Although this is a narrative review and this is not strictly required, it may be helpful to describe your method for literature search.
Answer: Thank you for taking the time to review this manuscript and for your comments and ideas that will improve our manuscript. Please find the detailed responses below and the corresponding corrections highlighted in the re-submitted files. Note that some sections highlighted in the revised manuscript are in response to the other reviewers.
A methodology section has been added on lines 65 to 72 and reads: Publications were selected if they were in English and reported original research or reviewed literature on asthma or gut microbiota in horses. Studies in humans, laboratory animals, and ex vivo studies were included if the results were relevant. The following keywords were used to conduct most of the literature searches in different databases (Pubmed, ScienceDirect, Scopus): mild to moderate and severe equine asthma, inflammatory airway disease, heaves, recurrent airway obstruction, equine gut microbiota, equine gut microbiome, equine gut dysbiosis, equine probiotics, gut-lung axis, asthma and dysbiosis. The most recent literature search was carried on December 2023.
Given the United States National Institute of Health’s policy on considering sex as a biological variable, I am wondering if you can comment on the role of sex differences in the gut-lung axis and equine asthma. In humans, young boys are more likely than girls to develop asthma, but after puberty females are more likely than males to develop asthma. There is evidence of sex differences in the microbiome. Are the same patterns found in horses? Have sex-differences related to the gut-lung axis been studied in the horse or is this an area where more research is needed?
Answer: This is a very interesting point. The reason for this is likely multifactorial but sex hormones seem to contribute as testosterone seems to have a protective against asthma. If we extrapolate from what we know in human medicine, we can hypothesize that because most males are geldings and have lower testosterone levels, the protective effect of being a male/having more testosterone is lost and therefore, no differences between males and females are observed in horses. To our knowledge, the only study investigating the differences between sex and microbiome in horses is a study by Mshelia et al. They found that there were significant differences in the fecal microbiome of mares and stallions, but unfortunately, there were no geldings in this study. In mice, there is evidence that microbiome contributes to sex differences observed in ozone-induced airway hyperresponsiveness in mice because no differences between males and females are observed if mice are treated with antibiotics to ablate gut microbiome (Cho et al, 2019). In humans, airway microbiome seems to differ by sex and there is an association between airway microbial markers, asthma and sex (Chen et al, 2020). However, to our knowledge, there is no sex-specific patterns in gut microbiota in humans associated with asthma. A few sentences with the appropriate references were added lines 310 to 326 to briefly address this point.
Line 143: Is there any evidence in racehorses that strenuous exercise alters the GI blood flow and tight junctions creating a leaky gut syndrome and that bacterial translocation from the GI to the respiratory tract is part of the asthma disease process in racehorses?
Answer: Again, thank you for introducing this interesting notion. Lines 282 to 286 now read: Because moderate exercise following transport increases intestinal permeability and systemic inflammation biomarkers (serum amyloid A and lipopolysaccharide) in horses, it can be hypothesized that bacterial translocation from the gastrointestinal tract to the respiratory tract could result in lower airway inflammation and/or be involved in asthma pathophysiology [85].
Line 449: Given the findings of the study of Schoster et al (PMID:25903509) where probiotic administration to healthy foals resulted in a greater incidence of diarrhea requiring treatment than in control foals, perhaps a word of caution is necessary regarding probiotic administration to foals?
Answer: This was added in lines 513 to 517: It is important to note that probiotic administration in foals is considered generally safe, but it can lead to adverse effects, such as an increased incidence of diarrhea requiring veterinary intervention [131]. Therefore, the use of probiotics in foals to alter the gut microbiota requires further investigation.
Line 471: I wonder if you could take the conclusions a bit further and say that the horse is a potential model for human asthma over the lifetime. With a focus on large scale data integration of longitudinal equine health records from foaling to adulthood, important questions related to the effects of illness and antimicrobial administration during the so-called “window of opportunity” on the later development of equine asthma could be investigated. And if interesting patterns emerge, maybe the horse could serve as a model for administration of pre-, pro-, and postbiotics during the vulnerable “window of opportunity” to mitigate the risk of developing asthma later in life.
Answer: This greatly improves the conclusion of the review. Lines 537 to 543 now read: The horse has the potential to serve as a model for human asthma across the lifespan. By focusing on large-scale data integration of longitudinal equine health records from foal to adulthood, researchers could investigate key aspects of the effects of disease and antimicrobial administration during the "window of opportunity" on equine asthma susceptibility. If promising trends emerge, the horse could serve as a model for the microbiota manipulation during the susceptible "window of opportunity" to reduce the risk of developing asthma in adulthood.
Reference [33]: Please provide complete citation.
Answer: This has been addressed. Thank you.